# Mitigating Modal Imbalance in Multi-Modal Reasoning

**Chen Henry Wu,**\*  **Neil Kale,**\*  **Aditi Raghunathan**
Carnegie Mellon University
{chenwu2,nkale,aditirag}@cs.cmu.edu

## Abstract

Foundation models (FMs) deployed in real-world tasks such as computer-use agents must integrate diverse modalities. How good are FMs at performing *joint reasoning*, simultaneously reasoning over multiple modalities, especially when the modalities interact and relate to each other to form *cross-modal context*? To better understand this problem, we study FMs on *cross-modal conflicts*: scenarios where conflicting evidence is presented across modalities. This allows us to examine whether FMs prioritize one modality over another or reason jointly to reconcile the conflict. Our experiments reveal that FMs can recognize conflicts in *unimodal contexts*, composed of a single modality, 90% of the time, but the ratio falls as low as 3% when evidence is split across modalities – similar observations hold in *cross-lingual contexts*, composed of multiple languages. We trace this failure to *cross-modal attention imbalance*, showing that FMs exhibit extreme asymmetry in attention scores, disproportionately prioritizing certain modalities. We show that cross-modal attention imbalance does not go away by simply scaling up multimodal or multilingual datasets blindly, since they lack training examples that explicitly require cross-modal reasoning. We demonstrate that even a simple and scalable method of explicitly combining multiple modalities within each training instance significantly reduces attention imbalance. Reduced attention imbalance directly translates to improved downstream performance on several vision-language benchmarks. Our findings underscore the importance of systematically addressing cross-modal contexts to build reliable foundation models.

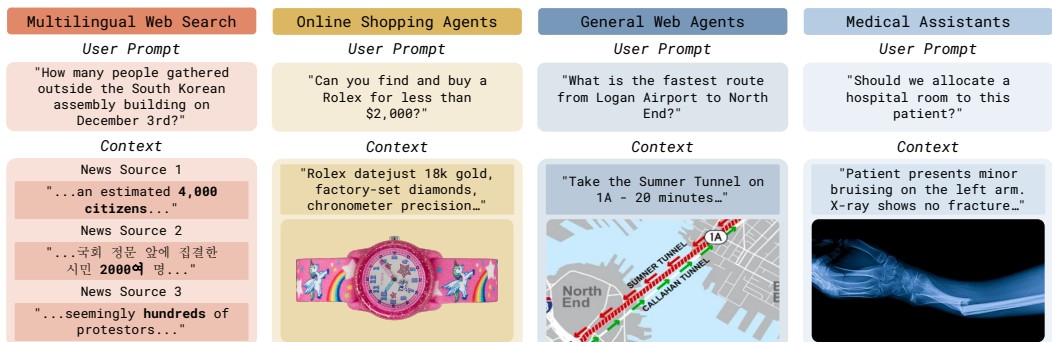

Figure 1: FM-based agents need to reason over diverse modalities, such as multilingual news, online shopping websites, maps, and EHR records. Failure to handle cross-modal context can result in consequences including misinformation (orange), purchasing a scam (yellow), misdirection (blue), or even providing the wrong medical treatment (light blue).

## 1 Introduction

Recent advances in foundation models (FMs; OpenAI, 2024a; Gemini, 2024; Anthropic, 2024b) have enabled their deployment in increasingly complex tasks that require reasoning

---

*Equal contributions

over diverse information sources. From autonomous web browsing (Adept, 2022; Anthropic, 2024a; OpenAI, 2024b) to AI-driven research assistants (Perplexity, 2024; Sakana, 2024), FMs are now tasked with reasoning jointly over multiple domains such as text, images, code, and structured data.

However, existing work indicates that FMs fall short when handling inputs from non-textual modalities. For example, some studies demonstrate that FMs answer visual questions primarily based on language priors, disregarding actual visual inputs (Winterbottom et al., 2020a; Niu et al., 2020; Lin et al., 2024); others illustrate scenarios where models hallucinate objects absent from the image during open-ended generation tasks (Sun et al., 2023). Yet, it remains unclear whether this behavior originates primarily from a context-parametric gap (Goyal et al., 2024) or a modality gap (Liang et al., 2022).

In this work, we specifically focus on the capability of FMs to reason across modalities when all necessary information is explicitly provided in the input context and does not conflict with parametric knowledge. This setup is especially relevant for FM agents and assistants that need to interpret up-to-date information unavailable within their parametric knowledge – such as web pages combining images, multilingual text, and embedded scripts (Figure 1). By designing scenarios that isolate cross-modal reasoning from parametric knowledge retrieval, we directly assess how effectively these models reason over multiple modalities.

As a clean and concrete test case for this, we create *cross-modal conflict* datasets where each modality provides contrasting evidence. This allows us to examine whether FMs prioritize one modality over another or reason jointly to reconcile the conflict. Joint reasoning means relating multiple pieces of information and identifying insights that can only be obtained from their interaction, not from any single piece alone. In our setup, joint reasoning over the contrasting evidence ideally identifies that there is no clear answer since the evidence is in direct conflict. Our experiments reveal a striking gap: while FMs perform well in *unimodal* settings (e.g., text-text or image-image), their ability to detect conflicts deteriorates significantly by up to 65% in *cross-modal* contexts (e.g., text-image). Moreover, this degradation extends to multilingual scenarios, where monolingual performance (e.g., English-English or Chinese-Chinese) is significantly better than multilingual performance (e.g., English-Chinese).

We investigate what drives this behavior, where state-of-the-art models exclusively rely on evidence from one modality rather than jointly reasoning. First, we observe that it is not simply a consequence of models being weak in one modality (§3.4). VLMs detect conflicts between multiple images as easily as conflicts between multiple texts (Figure 3). This extends to multilingual settings – conflicts between multiple Chinese texts are detected as often as in English.

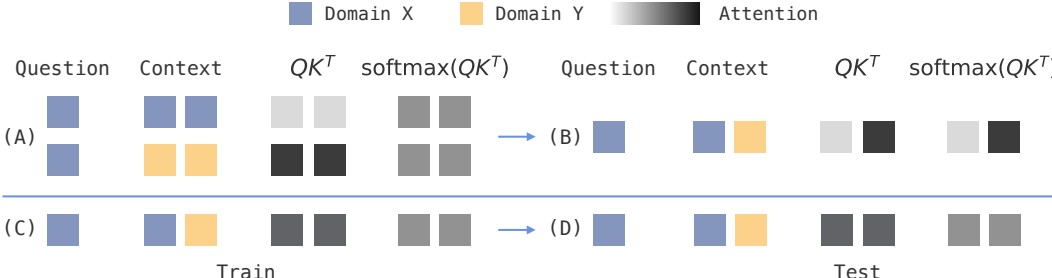

Figure 2: **An illustration for cross-modal attention imbalance.** In unimodal contexts (A), different domains show balanced normalized attention (softmax($QK^\top$)) despite divergent pre-softmax logits ($QK^\top$). Cross-modal contexts (B) expose cross-modal attention imbalance – normalization fails to mitigate logit-level imbalance. Instance-level modality mixing (C) resolves this by training models to intrinsically balance attention logits across modalities.

We hypothesize that the gap between unimodal and cross-modal conflict detection is because of *cross-modal attention imbalance*: an extreme asymmetry in attention scores, where FMs disproportionately prioritize certain modalities. We validate our hypothesis by finding that

manual attention reweighting vastly shifts the model towards joint reasoning rather than relying on one modality over another. Manual attention reweighting also directly improves downstream performance on a variety of hard vision-language benchmarks (§4.1).

We investigate how to correct cross-modal attention imbalance. The problem is not resolved by simply adding more training data in each modality (§5.1). As illustrated in Figure 2, when the cross-modal attention scores are imbalanced, different modalities have different pre-softmax logits ($QK^\top$). However, after normalization, unimodal contexts show balanced normalized attention (softmax($QK^\top$)) and their performance remains stable. So, fine-tuning on either modality separately does not reduce attention imbalance.

Current instruction-tuning datasets do not involve joint reasoning over multiple modalities. This is a known problem — curating data for non-textual modalities is expensive (Liu et al., 2023; Dai et al., 2023). It is infeasible to curate large amounts of joint reasoning data on top of the instruction data from each modality. However, we hypothesize that correcting for cross-modal attention imbalance is already sufficient to promote joint reasoning. A simple and scalable way to do this is to simply concatenate instructions from multiple modalities within the same context. In other words, we can repurpose existing datasets with this twist to greatly improve cross-modal joint reasoning in FMs.

In summary, we uncover a new fundamental gap between modalities – in terms of how they are processed in context. We demonstrate that state-of-the-art models fail in a simple cross-modal reasoning task of handling conflicting evidence from multiple modalities. We trace this failure to an imbalance in attention weights across modalities that can be addressed simply by mixing existing instruction data to create cross-modal instructions. Our findings also generally highlight the need for training paradigms that mirror the real-world complexity faced by FMs.

## 2 Related Work

**Modality Gap in Foundation Models**   FMs are known to fall short when handling inputs from low-resource modalities. For example, some studies demonstrate that FMs answer visual questions primarily based on language priors, disregarding actual visual inputs (Winterbottom et al., 2020a; Niu et al., 2020; Lin et al., 2024); others illustrate scenarios where models hallucinate objects absent from the image during open-ended generation tasks (Sun et al., 2023). Yet, it remains unclear whether this behavior originates primarily from a context-parametric gap (Goyal et al., 2024) or a modality gap (Liang et al., 2022).

Modern vision-language models (VLMs) embed text and images into a shared embedding space (Radford et al., 2021; Jia et al., 2021). The modality gap is characterized as separation between the embeddings of different data modalities (Liang et al., 2022) which hurts performance on visual question answering and classification tasks (Guo et al., 2023; Winterbottom et al., 2020b). Several explanations have been proposed, including inductive bias of encoders and disuniformity of contrastive loss (Fahim et al., 2024). A similar phenomenon persists in multilingual FMs too (Nigatu et al., 2023; Chang et al., 2022).

**Multilingual and Multimodal Instruction Tuning**   Multilingual and vision-language models employ specialized pre-training data (Ustun et al., 2024; Li et al., 2024a) and instruction-tuning datasets (Li et al., 2023; Liu et al., 2023; Antol et al., 2015) to improve performance on underrepresented modalities. In general, however, these models are designed for unimodal performance and they do not saturate large cross-modal benchmarks like MMMU and ScienceQA that require simultaneously reasoning over data in multiple modalities (Yue et al., 2023; Lu et al., 2022).

Several fine-tuning approaches have been suggested to improve cross-modal reasoning in FMs. For example, X-InstructBLIP claims that training on individual modalities can result in emergent cross-modal reasoning (Panagopoulou et al., 2023). In our analysis, however, we find strong evidence that this is not always possible.

**Related Approaches for Cross-Modal Reasoning**   Another common approach to improve cross-modal performance is to mitigate language bias, or over-dependence on language priors (Niu et al., 2020). This approach prevents the model from ignoring images due to parametric knowledge about the question, but does not counteract bias within the context towards textual evidence over image evidence.

Other more specialized approaches include aligning individual entities between modalities (Lin et al., 2022) or learning sparse feature representations that rely less on language priors (Guo et al., 2021). In addition, Li et al. (2024b) propose a fine-tuning approach that is similar to our instance-level mixing strategy; however, they focus only on textual data, whereas we focus entirely on mixing modalities.

**Knowledge Conflicts in Cross-Modal Contexts**   Even in unimodal settings, FMs sometimes fail to identify when they encounter conflicting information (Xu et al., 2024; Xie et al., 2023). In these unimodal scenarios (e.g., outdated facts), there is evidence that FMs have the ability to identify when they don't know an answer (Kadavath et al., 2022; Yin et al., 2023). In addition, mitigation strategies like prompting (Zhou et al., 2023), pretraining (Li et al., 2022), and reweighting neurons (Shi et al., 2024) are known to improve detection but remain limited to specific unimodal contexts. Existing instruction-tuning solutions for conflict detection (Wang et al., 2023) rely heavily on curated conflict-specific datasets. Furthermore, existing benchmarks focus on textual knowledge conflicts (Su et al., 2024).

Notably, prior work largely overlooks knowledge conflicts between multiple modalities. Liu et al. (2024b) benchmarks cross-modal conflicts, but focuses on context-parametric conflicts between images and the model's pretrained knowledge. Zhu et al. (2024) also studies cross-modal conflicts, but focuses on parametric conflicts between the language model and visual encoder components of a multimodal FM. To the best of our knowledge, knowledge conflicts have not previously been used to study cross-modal reasoning in a controlled setting.

# 3   Stress-testing cross-modal reasoning

## 3.1   Formulation

We study the free-form generation from a FM. The FM takes a context $C$ and a question $Q$ as input and samples a response $y \sim \text{FM}(C, Q)$. The context $C$ has an in-context knowledge conflict, i.e., $C$ contains two subsequences, $C_1$ and $C_2$, that support contradictory answers to $Q$. We consider $(C, Q)$ to be a unimodal conflict if $C_1, C_2 \in M_1$ and a cross-modal conflict if $C_1 \in M_1, C_2 \in M_2$ where $M_1, M_2$ are distinct modalities. This allows us to examine whether FMs prioritize one evidence over another or reason jointly to reconcile the conflict. Given a set of context-question pairs $\mathcal{D} = \{(C_i, Q_i)\}_{i=1}^N$, we define the *conflict detection rate* as the proportion of samples that are mentioned to contain conflicts. We used GPT-4o as the evaluator (see prompts in §C.1). To isolate the context-based reasoning independent of the parametric bias, we focus on tasks that depend on the context and cannot be solved with the parametric knowledge alone.

## 3.2   Data curation

We construct two datasets: a *cross-modal question answering (CMQA)* dataset and a *cross-lingual question answering (CLQA)* dataset, each with controlled variations in context. Examples from both data sets are given in §B.

**Cross-modal question answering (CMQA)**   The multimodal question answering dataset is constructed over both image and text based on the VQA-v2 dataset (Goyal et al., 2016). Each sample in VQA-v2 consists of an image $V$, a question $Q$, and 10 candidate answers. In total, we subsample 500 triples of image, question, and answer $(V, Q, A)$ from the dataset.

For each triplet, we prompt GPT-4o to generate a text description $\overline{T}$ that does not agree with the image $V$ regarding the question $Q$, and the answer $\overline{A}$ based on $\overline{T}$. Given the image $V$, the

text description $\overline{T}$, and the question $Q$, the FM should report a conflict as $A$ is contradictory to $\overline{A}$. We name this dataset $\{(V, \overline{T}, Q, A, \overline{A})\}$ as Text-Image. For each image $V$, we prompt GPT-4o to generate a description $T'$ that agrees with the image regarding the question $Q$. We name $\{(T', \overline{T}, Q, A, \overline{A})\}$ as Text-Text. For each $\overline{T}$, we prompt DALL-E 3 (Betker et al., 2023) to generate an image $\overline{V}$. We name $\{(V, \overline{V}, Q, A, \overline{A})\}$ as Image-Image.

**Cross-lingual question answering (CLQA)** We create a dataset of question answering over synthetic news paragraphs about fictitious events (so the FM cannot use parametric knowledge to answer the questions). We use GPT-4o to generate 400 topics. For each topic, we prompt GPT-4o to generate: (1) a synthetic news paragraph $P_E$ in English which has not appeared in reality, a question $Q$ in English, and an answer $A$ based on the paragraph, and (2) synthetic news paragraph $\overline{P}_C$ in Chinese that does not agree with the English one $P_E$ regarding the question $Q$, and an answer $\overline{A}$ based on the Chinese one.

Given the two news paragraphs $P_E$ and $\overline{P}_C$ and the question $Q$, the FM should reason over both since $A$ is contradictory to $\overline{A}$. We name this cross-modal dataset $\{(P_E, \overline{P}_C, Q, A, \overline{A})\}$ as English-Chinese. We then derive several monolingual variants of different language combinations via (back-) translation. For each paragraph $\overline{P}_C$ in Chinese, we back-translate it into English $\overline{P}'_E$. We name $\{(P_E, \overline{P}'_E, Q, A, \overline{A})\}$ as English-English. For each English paragraph $P_E$, we translate it into Chinese $P'_C$. We name $\{(P'_C, \overline{P}_C, Q, A, \overline{A})\}$ as Chinese-Chinese. Similarly, we test other cross-lingual variants where Chinese is replaced with low-resource languages such as Turkish or Icelandic.

### 3.3 Experimental setup

We evaluate a range of state-of-the-art (multimodal) FMs on our conflict detection tasks. Multimodal FMs can be applied to both CLQA and CMQA, while text-only FMs can only be applied to CLQA. For text-only FMs, we use Llama-3 and Llama-3.1 (Meta, 2024), Gemma-2 (Riviere et al., 2024), and Aya-23 (Ustun et al., 2024). For multimodal FMs, we use GPT-4o (OpenAI, 2024a), LLaVA-NeXT (Li et al., 2024a), and Cambrian (Tong et al., 2024). We provide the prompts we use in §C.2.

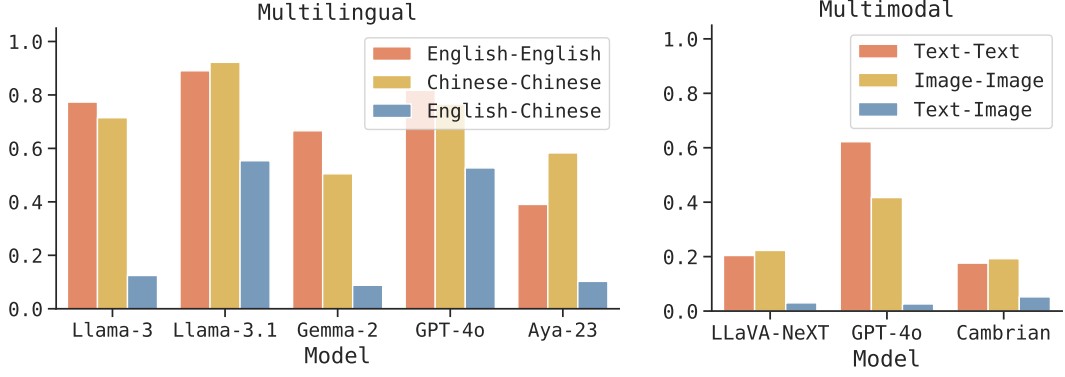

Figure 3: FMs are worse at reasoning over cross-modal contexts than unimodal contexts.

### 3.4 Results

Figure 3 shows the performance of the FMs on our CLQA and CMQA datasets. We see that the conflict detection performance is significantly lower with the cross-modal contexts than with the unimodal contexts.

For CLQA (Figure 3 left), we the performance on English-English is comparable to Chinese-Chinese, and both are far better – up to 5x – than English-Chinese. This shows that the lower performance in the multilingual setting is not due to the limited general capability

of the FM in Chinese. Also, recall that the questions are always in English, including in the Chinese-Chinese setting, so the lower performance with multilingual contexts is neither due to the language barrier between English and Chinese. Experimental results for Turkish and Icelandic are similar to those for Chinese, so we put them in §E for conciseness. We see similar trends on the CMQA task (Figure 3 bottom) – the performance with unimodal contexts (Text-Text and Image-Image) is far better than the performance with cross-modal contexts (Text-Image) for all FMs.

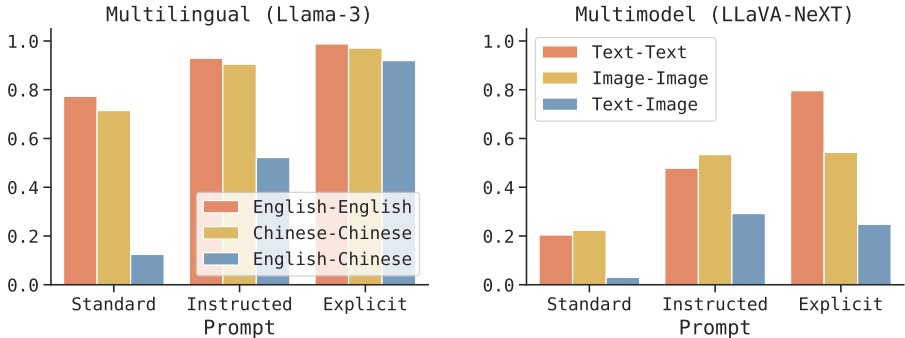

Figure 4: Ablation studies on the prompt. FMs are worse at reasoning over cross-modal contexts than unimodal contexts. See the text for details of each prompt.

In addition to the prompts above (denoted as Standard in Figure 4), which do not assume intervention from the user, we also test other prompts that encourage FM to detect the conflict. Specifically, we explore two types of prompts: (1) add an instruction that tells the FM to report the conflict if it finds any (denoted as Instructed in Figure 4); (2) embed the question into a yes-no question: "Would the answers to the question '{Q}' be the same based on the paragraphs in the context?" (denoted as Explicit in Figure 4). In Figure 4, we see that, although the overall conflict detection performance improves, the trend is similar to Figure 3 – the conflict detection performance is lower in the cross-modal contexts than in the unimodal contexts. In the next section, we explore why this is the case and try to improve this practically.

**Summary.** State-of-the-art FMs fail in a simple cross-modal reasoning task of handling conflicting evidences in multiple modalities.

## 4 Cross-modal attention imbalance

### 4.1 Attention imbalance in FMs

To investigate the mechanisms underlying this failure, we probe the context contribution in FMs. Most state-of-the-art FMs are autoregressive – at each step, the FM predicts the next token based on the context so far. For architectures like Transformers (Vaswani et al., 2017), the representation at each step can be decomposed into a linear combination of the contributions of each span of context. For example, in a Transformer FM, the output of an attention head in a layer at step $t$ is defined as:

$$\boldsymbol{a}_t = \boldsymbol{W}_O \sum_{j=1}^{t} w_{t,j} \boldsymbol{v}_j, \tag{1}$$

where $\boldsymbol{v}_j$ is value output of the $j$-th token in the context, $w_{t,j}$ is the attention weight from the $t$-th token to the $j$-th token, and $\boldsymbol{W}_O$ is the output projection matrix. We can group tokens in the context based on their domain: $\mathcal{C}_k$ contains all token indices of the $k$-th group. Based on

this, we can rewrite the above equation as:

$$a_t = \sum_{k=1}^{K} \left( \sum_{j \in \mathcal{C}_k} w_{t,j} W_O v_j \right) := \sum_{k=1}^{K} u_k. \tag{2}$$

The term $u_k$ is a vector that the $k$-th context writes to the residual stream at step $t$. It shows that the context representation is a linear combination of each context's contribution.

We hypothesize that the context contribution *in the task-relevant subspaces* is imbalanced in the cross-modal contexts, making the FM more likely to rely on the dominant context instead of doing conflict detection. In Figure 2, we illustrate our mental model of attention imbalance. In unimodal contexts (A), different domains show balanced normalized attention (softmax($QK^\top$)) despite divergent pre-softmax logits ($QK^\top$). Cross-modal contexts (B) expose cross-modal attention imbalance – normalization fails to mitigate logit-level imbalance. Instance-level modality mixing (C) resolves this by training models to intrinsically balance attention logits across co-occurring domains.

To demonstrate attention imbalance, we compute the average norm of $u_k$ for each context, averaged over all layers and attention heads.[1]  Figure 5 shows that, for cross-lingual, the English context contributes more than the Chinese context; for cross-modal, text contributes more than images.

To test if there is a *causal* relationship between attention imbalance and cross-modal reasoning, we causally intervene the contribution of a context $\mathcal{C}_k$ by adding a small constant $\epsilon$ to its unnormalized attention score. Formally, denote the normalized attention weights at step $t$ as $w_t := [w_{t,1}, \ldots, w_{t,t}]^\top$. We manipulate the attention weights as follows:

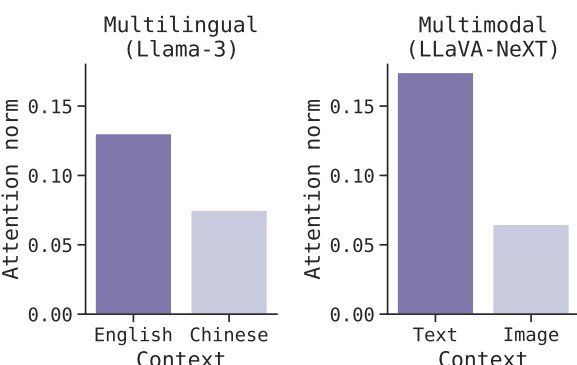

Figure 5: Cross-modal attention imbalance. English has larger attention than Chinese and images.

$$\text{Manip}(w_t) = \text{softmax}\left( \log w_t + \epsilon \mathbf{1}_{\mathcal{C}_k} \right), \tag{3}$$

where $\mathbf{1}_{\mathcal{C}_k}$ is a vector with 1's on all the $\mathcal{C}_k$ context positions and 0's otherwise.

Figure 6 shows that the conflict detection performance indeed improves after attention manipulation. In the cross-lingual setting, the absolute improvement is up to 43% (relative by 5x). In the cross-modal setting, we observe a smaller yet significant gain of 18%. As a side observation, we find that attention manipulation can help the unimodal context as well: we find that FMs exhibit primacy bias and tend to rely more on the context that appears first. By increasing the attention weights on later context, we also further improve the conflict detection performance on unimodal context.

## 4.2 Verification on other datasets

To verify the generality of our findings, we applied the attention manipulation to three benchmarks that require substantive visual reasoning. Popular datasets such as MMMU show only marginal performance differences between text-only and vision-language models, which limits their diagnostic value (Fu et al., 2024). In contrast, we use two more challenging datasets: BLINK (Fu et al., 2024) and SAT (Ray et al., 2024).

---

[1]We note that $u_k$ averaged over all layers and attention heads should be viewed as a proxy of what we want to measure, i.e., context contribution *in the task-relevant subspaces*. We further discuss this in §D. For this reason, we do not argue that the norms of different $u_k$ should be the same to achieve the best conflict detection performance.

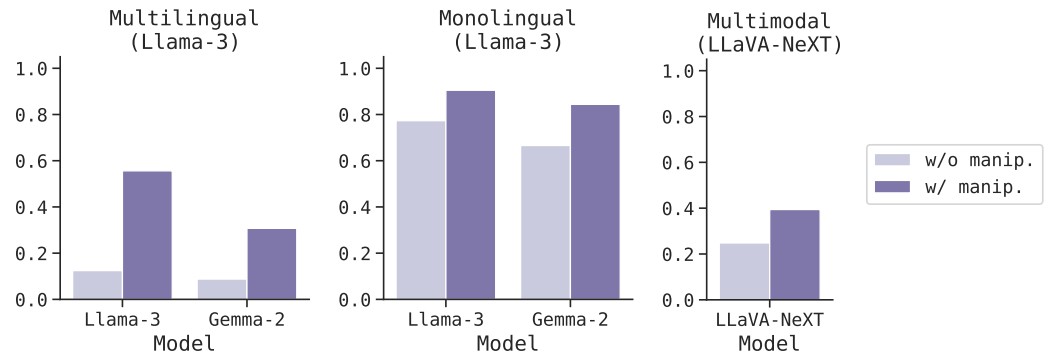

Figure 6: Cross-modal attention imbalance has a causal effect on cross-modal reasoning: we apply a fixed attention bias to increase the attention over the context with a smaller attention output norm, and see that this improves the conflict detection performance. We use the `Standard` prompt for cross-lingual and monolingual settings, and `Explicit` prompt for the cross-modal setting.

We apply the attention manipulation to the LLaVA-NeXT 8B model, where we sweep the manipulation strength from -1 (less visual attention) to 1 (more visual attention). Figure 7 shows that attention manipulation achieves an absolute accuracy gain of 1% and 2%. These results indicate that reducing attention imbalance between modalities is especially effective when visual reasoning is indispensable.

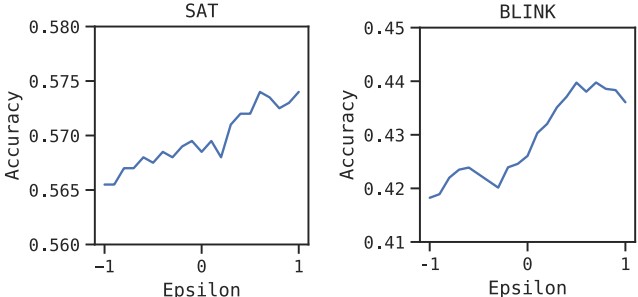

Figure 7: The performance on cross-modal reasoning datasets improves as we increase the strength of attention manipulation.

> **Summary.** Cross-modal attention imbalance has a causal negative effect on FMs' cross-modal reasoning capability.

## 5 How does training affect cross-modal attention imbalance?

### 5.1 Dataset-level modality mixing does not help

We begin by noting that most state-of-the-art FMs today are trained on highly diverse corpora, spanning a wide range of domains and multiple languages (Meta, 2024; Riviere et al., 2024). More surprisingly, as we observe in Figure 3, Aya-23, a FM specifically optimized for multilingual capabilities, performs no better than other FMs with multilingual contexts. This suggests that simply training FMs on diverse modalities does not, by itself, ensure good cross-modal reasoning.

To reinforce this, we run two instruction-tuning experiments. First, we finetune Llama-3 on mixed English and Chinese instruction tuning datasets (we call this strategy **dataset-level modality mixing**) and see if this improves conflict detection in the cross-lingual `English-Chinese` setting. Specifically, we use the English and Chinese subsets of Bactrian-X (Li et al., 2023), a multilingual instruction-tuning dataset containing 67k samples in each language. Similarly, we finetune Qwen-2.5-VL on mixed text and visual instruction tuning datasets (**dataset-level modality mixing**) and see if this improves conflict detection in the cross-modal `text-image` setting. In this experiment, we use the visual instruction data from Liu et al. (2024a) and the English subset of Bactrian-X. In Figure 8, we see that *dataset-level modality mixing* offers minimal gains in alleviating cross-domain attention imbalance. This

motivates us to understand why diverse, multimodal data is not enough to close the gap between unimodal and cross-modal contexts.

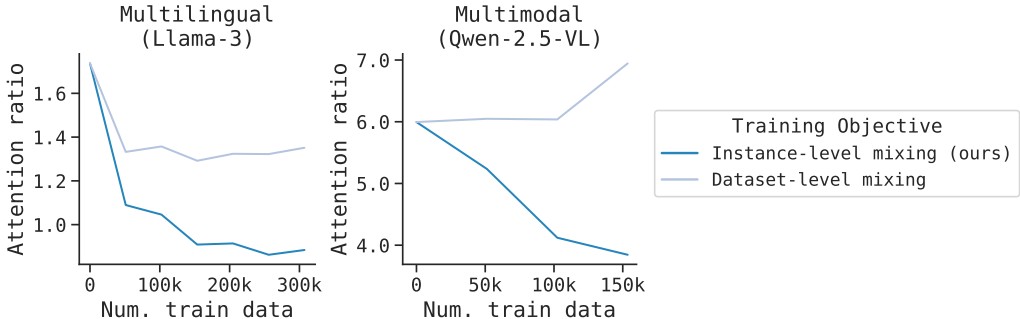

Figure 8: Finetuning on instance-level mixed data (dark blue) reduces cross-modal attention imbalance significantly more than fine-tuning on dataset-level mixed data (gray).

## 5.2 Instance-level modality mixing

We have shown that standard cross-modal instruction tuning (e.g. having both English and Chinese examples in the data, or having both text and visual instruction tuning examples) fails to improve cross-modal attention imbalance. We hypothesize that the gap in unimodal and cross-modal contexts arises because mixing datasets does not expose models to instances requiring cross-domain reasoning *within the same context*. Without instance-level modality mixing between modalities, the pre-softmax attention scores for one domain could be hugely different from that of another domain, without changing the normalized attention scores on each domain (Figure 2). To address the lack of instance-level modality mixing between modalities, we propose a simple and scalable method of explicitly combining multiple modalities within each training instance. Here is an illustration of our input and output format in the cross-lingual setting:

```
Input:
<Chinese instruction> <English instruction>
Reply to both user instructions.
Output:
<Chinese response> <English response>
```

In the cross-modal setting we have the input and output as:

```
Input:
<text instruction>
<image> <image-related instruction>
Reply to both user instructions.
Output:
<text response> <image-related response>
```

To verify the benefit of instance-level modality mixing, we use the same data from § 5.1 but mix them at the instance level instead of at the dataset level. In Figure 8, we report the attention imbalance of model checkpoints for instance-level modality mixing and the dataset-level modality mixing baseline in §5.1. In the cross-lingual setting, instance-level modality mixing reduces attention imbalance between modalities by 4×. In the cross-modal setting, it reduces attention imbalance by 34%. In Figure 9, we report the performance of model checkpoints for *instance-level modality mixing* and the baseline in §5.1 (*dataset-level modality mixing*). In the cross-lingual setting, instance-level modality mixing boosts conflict detection by 37%, much greater than dataset-level modality mixing. In the cross-modal setting, instance-level modality mixing improves conflict detection by 2×.

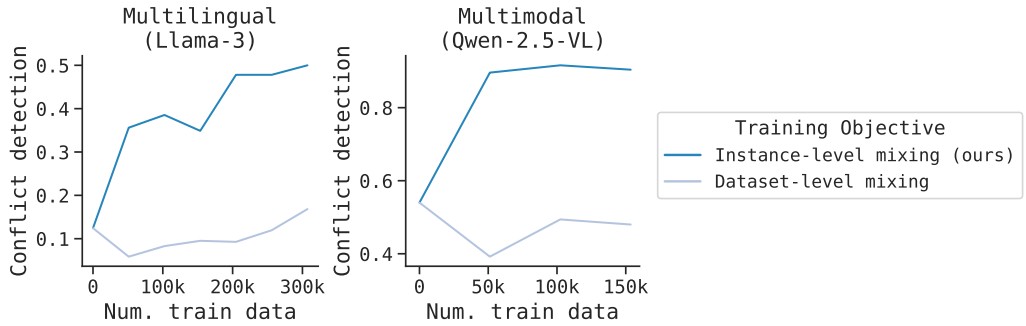

Figure 9: Finetuning on instance-level mixed data (dark blue) improves cross-modal conflict detection largely more than fine-tuning on traditional dataset-level mixed data (gray).

We highlight that instance-level modality mixing is more scalable than directly finetuning the FMs on the knowledge conflict detection task itself, as it does *not* require any explicit conflicts within the instructions, which could be costly to generate for diverse domains. Notably, the improvement in conflict detection does not come from training on the same task that we are testing on, but rather a general proxy for cross-modal attention balance.

**Summary.** Instance-level modality mixing mitigates attention imbalance and improves cross-modal reasoning, without requiring any additional data curation.

### 5.3 Verifying instance-level modality mixing on downstream tasks

Synthetic data must be validated against real-world complexity. Similar to manual attention rebalancing (§4.2), we find that instance-level modality mixing improves performance on several hard vision-language benchmarks. These experiments indicate that our proposed methods to improve conflict detection also enhance the overall reasoning capabilities of models, especially on datasets that require non-trivial reasoning over each domain. As demonstrated in Figure 10, fine-tuning on instance-level mixed data improves performance by 2.4% on HardBLINK, 2.4% on the static subset of SAT, 4.4% on the dynamic subset of SAT, and 1.2% on MMMU.

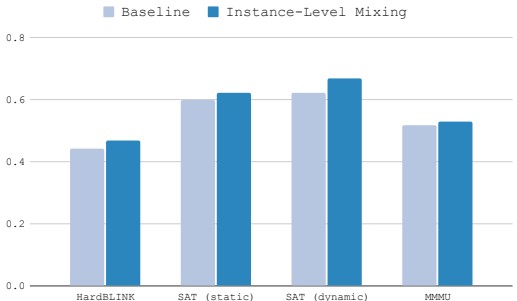

Figure 10: Finetuning Qwen-2.5-VL on instance-level mixed data improves performance on hard multimodal benchmarks.

## 6 Conclusions

We uncovered a new fundamental gap in how FMs process modalities in context. Through controlled datasets and experiments, we demonstrated that state-of-the-art models fail in a simple cross-modal reasoning task of handling conflicting evidences in multiple modalities. Our analyses trace the problem to *cross-modal attention imbalance*, an imbalance in attention weights across modalities. We showed that simply including multiple modalities in training (i.e., dataset-level modality mixing) has little gains, while explicitly mixing different modalities within each training sample (i.e., instance-level modality mixing) mitigates attention imbalance and substantially boosts conflict detection. Our results highlight the need for training paradigms that mirror the real-world complexity faced by models and for scaling methods that enable foundation models to balance cross-modal attention and reason on cross-modal contexts.

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

# A    Additional examples of conflict detection

We provide additional examples of FM failure in conflict detection over cross-modal contexts.

**Multilingual example**    In Figure 11, we provide an example of how an FM (GPT-4o) with web access can fail to acknowledge knowledge conflicts from multilingual news sources.

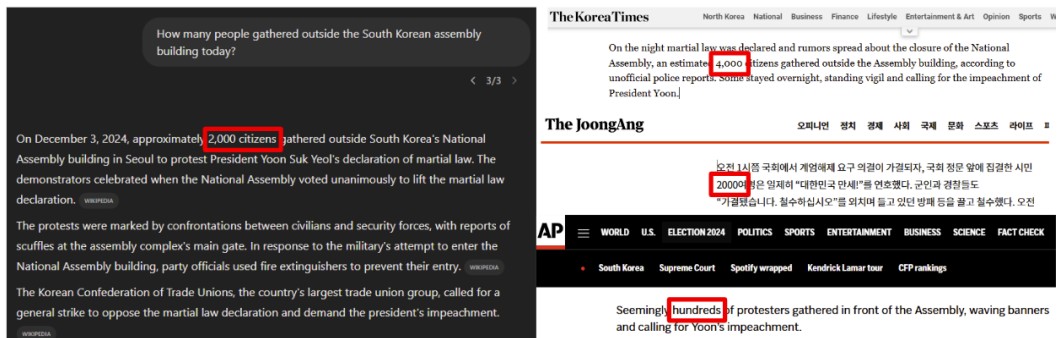

Figure 11:   An FM with web access can fail to acknowledge knowledge conflicts from multilingual news sources.  For example, GPT-4o reports the size of the protest outside South Korea's National Assembly on December 3, 2024 as 2,000 people, although different sources provide conflicting numbers of attendees.

**Cross-modal agent example**    In Figure 12, we provide an example of how an FM (GPT-4o) with web access can fail to acknowledge knowledge conflicts in multimodal product descriptions.

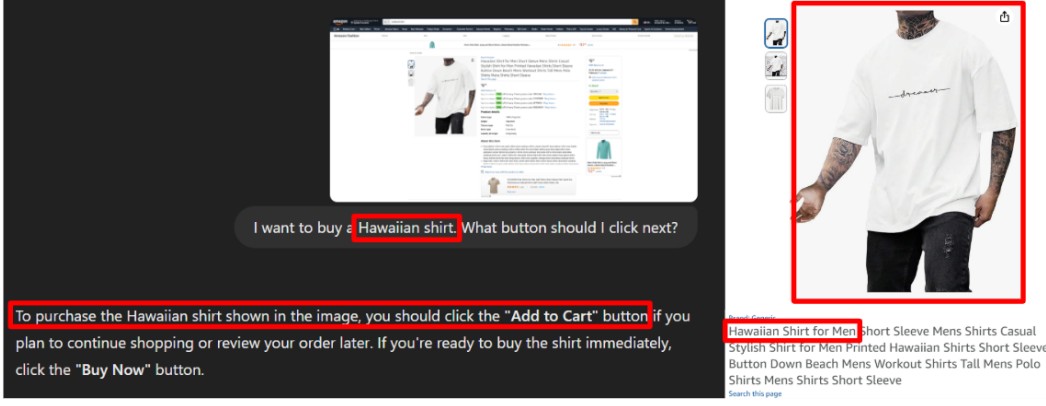

Figure 12:  A FM can fail to acknowledge knowledge conflicts in multiple modalities. For example, GPT-4o instructs the user to purchase an item labeled as "Hawaiian Shirt for Men" despite the image clearly depicting an ordinary t-shirt, not a Hawaiian shirt.

# B    Dataset Examples

Figure 13 shows an example English evidence, Chinese evidence, and question from our CLQA dataset. Figure 14 shows an example text evidence, image evidence, and question from our CMQA dataset.

In a recent breakthrough, researchers at the University of California have developed a new genetic modification technique that significantly boosts crop yield. This advancement, based on CRISPR-Cas9 technology, allows for more precise editing of plant genomes, enabling scientists to enhance growth rates and resistance to environmental stressors. Preliminary field tests conducted on corn and wheat in the San Joaquin Valley have shown promising results. The modified crops exhibited a robust increase in productivity, with yield improvements recorded at approximately 25\%. This development is expected to revolutionize agricultural practices by reducing the need for chemical fertilizers and pesticides, potentially lowering production costs and benefiting farmers globally. According to Dr. Emily Zhang, the lead scientist on the project, the technique is ready for wider application and could be instrumental in addressing food security challenges posed by a growing global population and climate change. The next steps involve scaling up production and collaborating with agricultural organizations to implement these genetically modified crops on a larger scale. However, some environmental groups have raised concerns about the long-term impacts on biodiversity and ecosystem balance, advocating for more rigorous testing before widespread adoption.

最近，加州大学的研究人员开发了一种新的基因改造技术，可以显著提高农作物产量。该技术基于CRISPR-Cas9技术，允许更精确地编辑植物基因组，从而增强生长速度和对环境压力的抵抗力。在圣华金谷进行的玉米和小麦初步田间试验显示出令人鼓舞的结果。经过改造的作物表现出生产力的显著提高，产量提高约为15%。这一发展预计将通过减少对化肥和农药的需求来革新农业实践，可能降低生产成本并使全球农民受益。项目负责人张艾米博士表示，该技术已准备好进行更广泛的应用，并可能在应对由全球人口增长和气候变化带来的粮食安全挑战中发挥关键作用。接下来的步骤包括扩大生产规模，并与农业组织合作，在更大范围内实施这些转基因作物。然而，一些环保团体对生物多样性和生态系统平衡的长期影响表示担忧，呼吁在广泛采用之前进行更严格的测试。

Q: What is the estimated percentage increase in crop yield due to the new genetic modification technique?

Figure 13: English evidence, Chinese evidence, and question from our CLQA dataset.

In the breathtaking expanse of a winter wonderland, the person is immersed in the art of snowboarding. As they glide effortlessly down the pristine slopes, their board carves precise arcs into the powdery snow, leaving behind a trail of skillful mastery. Clad in vibrant winter gear, they exhibit the perfect blend of agility and grace that is the hallmark of a seasoned snowboarder. Each twist and turn is a testament to their years of practice and passion for the sport. The snowy landscape stretches out endlessly, a canvas for the snowboarder's dynamic movements. With each jump and trick, they defy gravity, soaring briefly before landing with practiced ease. The sun shines brightly, reflecting off the snow and illuminating the snowboarder's path as they navigate the mountain with confidence. Every moment on the board is a dance with the elements, a thrilling experience that captivates both the participant and any fortunate observers. In this wintry realm, the snowboarder finds freedom and exhilaration in equal measure, making the most of every descent.

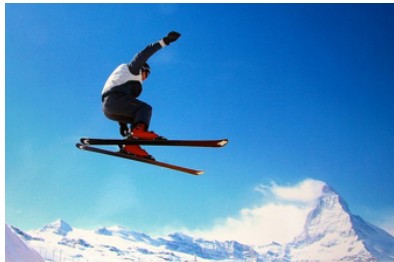

Q: What is the person doing?

Figure 14: Text evidence, image evidence, and question from our CMQA dataset.

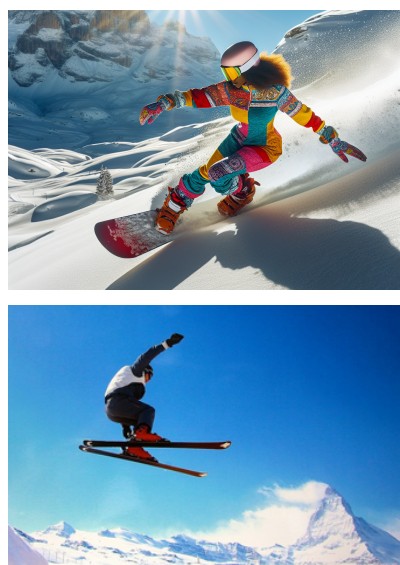

Q: What is the person doing?

Figure 15: Multi-image evidence and question from our CMQA dataset.

## C Prompts

### C.1 LM-as-a-Judge for Evaluating Conflict Detection

We use the following prompt for GPT-4o to evaluate if the model output is doing conflict detection.

```
I'll provide you with a question and a response by a model.

## Your task

Can you infer from the response that the pieces of evidence provided to the model
has contradictions within them (or that different pieces of evidence suggest
different answers)?
If yes, then contradict_score = 1, otherwise contradict_score = 0.

Examples of contradictions:
- two images describing different things, e.g., one image describes something,
while the other image describes something else;
- image and text describing different things, e.g., the image describes something,
while the text describes something else;
- two paragraphs giving different answers, e.g., one paragraph says something,
while the other paragraph says something else.

## Input

Question: {question}
Response: {response}

## Output requirements

Wrap your final judgement in triple backticks. Your judgement should look like
this:

```json
{{
    "contradict_score": 0 or 1
}}
```
```

## C.2 CMQA and CLQA Prompts

We prompt FMs with two pieces of context and a question, and sample a response from the FM. Here is the prompt we use for our CLQA task:

```
Here are some paragraphs.
<paragraph_1>
<paragraph_2>
Based on all the paragraphs, answer the question below. Reply in English.
<question>
```

Similarly, we use the following prompt for our CMQA task:

```
<image> <text>    // The order depends on the model.
Above are visual and textual descriptions of a scene.
Answer the question below.
<question>
```

## D   Additional experiments in attention imbalance

Recall that in §4.1 we demonstrate cross-modal attention imbalance with the average norm of $u_k$ for each context, averaged over all layers and attention heads. In this section, we elaborate on this by visualizing $u_k$ for each context in each layer and attention head.

Figure 16 visualizes the norm of $u_k$ for each layer and attention head in the multilingual setting, aggregated over all test samples. Figure 17 visualizes the norm of $u_k$ for each layer and attention head in the multilingual setting, aggregated over all test samples. We see that that the values over the Chinese/image context is generally smaller than those over the English/text context, especially in upper layers.

We note one important exception that is relevant to the footnote in §4.1, where we argue that $u_k$ averaged over all layers and attention heads should be viewed as a proxy of what we want to measure, i.e., context contribution *in the task-relevant subspaces*. Figure 16, Layers 11-14 are an exception, where the values over the Chinese context is higher – we argue that these layers are *not* in the task-relevant subspace (i.e., they activates on the Chinese context but does not improve the reliance on Chinese when answering the question). For this reason, we do not argue that the norms of different $u_k$ should be the same to achieve the best conflict detection performance.

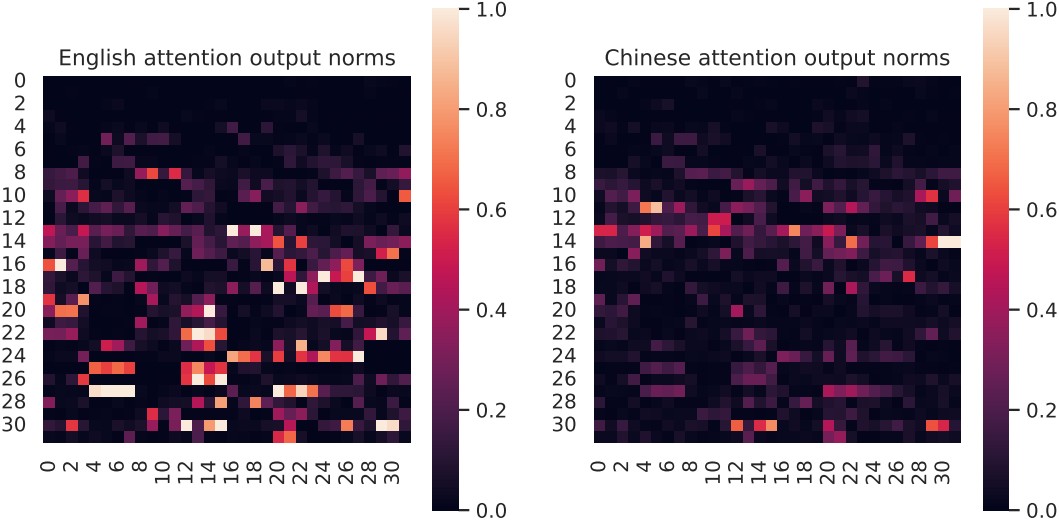

Figure 16: We visualized the norm of $u_k$ for each layer and attention head in the multilingual setting, aggregated over all test samples. We see that the values over the Chinese context is generally smaller than those over the English context, especially in upper layers. Notably, Layers 11-14 are an exception, where the values over the Chinese context is higher – we argue that these layers are *not* in the task-relevant subspace (i.e., they activates on the Chinese context but does not improve the reliance on Chinese when answering the question).

## E   Additional results on other languages

Figure 18 shows the results of conflict detection over cross-modal contexts containing Icelandic and Turkish.

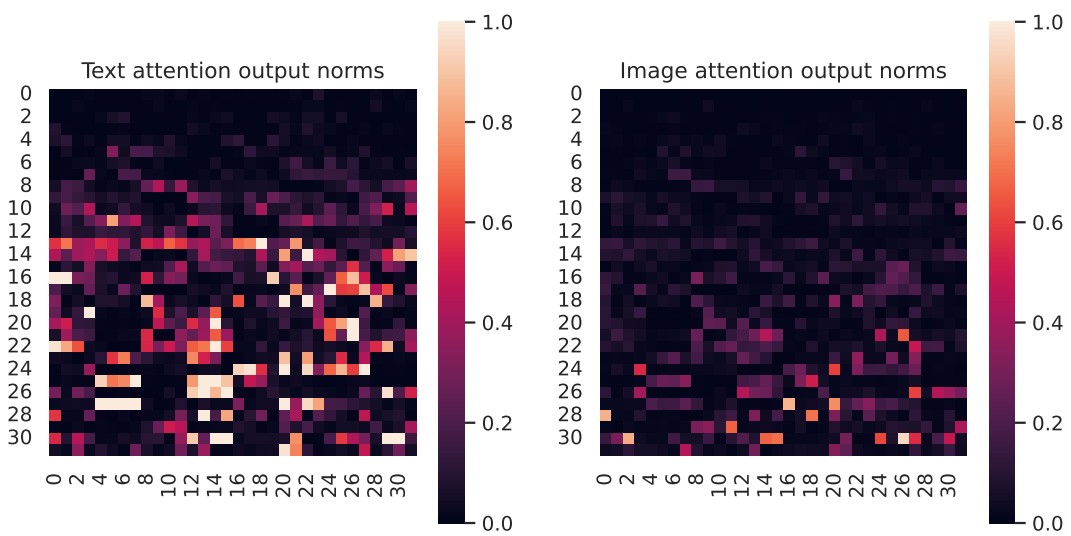

Figure 17: We visualized the norm of $u_k$ for each layer and attention head in the multimodal setting, aggregated over all test samples. We see that the values over the image is generally smaller than those over the text.

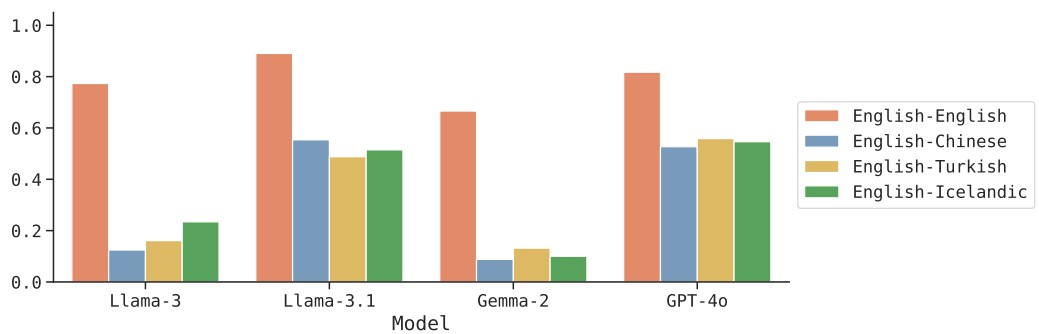

Figure 18: Conflict detection ratio over cross-modal contexts with Icelandic and Turkish.

