# OpenReview forum: "Mitigating Modal Imbalance in Multimodal Reasoning"
_colmweb.org/COLM/2025/Conference — COLM 2025_

### Official Review · Reviewer_878T · 2025-05-11

**Rating:** 7
**Confidence:** 4
**Ethics Flag:** 1

**Summary:**

This paper investigates a critical limitation of foundation models (FMs) in cross-modal reasoning, specifically their inability to effectively reconcile conflicting evidence across multiple modalities (e.g., text-image or multilingual contexts). The authors introduce controlled datasets for cross-modal and cross-lingual question answering tasks where conflicts are explicitly engineered. They find that while FMs perform well in unimodal settings (e.g., English-English or Text-Text), their performance drops significantly in cross-modal settings (e.g., English-Chinese or Text-Image), often failing to detect contradictions. Through detailed analysis, the authors identify a core issue: cross-modal attention imbalance , where certain modalities dominate the model’s attention, leading to biased reasoning. Finally, they propose a simple yet effective solution—instance-level modality mixing during training—which mitigates the imbalance and improves conflict detection.

**Reasons To Accept:**

1. The paper presents a strong empirical foundation. The authors design synthetic datasets (CMQA and CLQA) with explicit knowledge conflicts across modalities and languages. These datasets allow for clean evaluation of cross-modal reasoning without confounding factors from parametric knowledge. The experiments are comprehensive, covering state-of-the-art multimodal and multilingual models (GPT-4o, LLaVA-NeXT, Llama-3, etc.) and varying prompting strategies.
2.The theoretical analysis of attention mechanisms is rigorous and insightful. The decomposition of context contributions via attention heads and layers provides a principled way to quantify modality imbalance. The causal intervention experiments (adding bias to attention weights) offer compelling evidence that attention imbalance directly affects reasoning performance.

**Reasons To Reject:**

1. The representativeness of the synthetic datasets is limited . The authors use GPT-4o to generate cross-modal conflict examples, such as text-image or multilingual QA pairs. While this enables controlled experimentation, the resulting data lacks the complexity, ambiguity, and diversity found in real-world multimodal contexts. For instance, conflicts in real applications often arise from nuanced or implicit contradictions rather than explicit textual or visual mismatches.

2. The proposed method of instance-level modality mixing closely resembles the structure of the synthetic data used for evaluation . Specifically, the approach involves concatenating modalities (e.g., English + Chinese instructions) within the same context during training. However, this form of cross-modal concatenation mirrors the structure of the synthetic conflict examples used in the evaluation (e.g., presenting conflicting English and Chinese paragraphs side-by-side). As a result, the observed performance improvements may be partially attributable to the alignment between the training data format and the test data format, rather than a genuine improvement in cross-modal reasoning capability.

---

> ### Author Response · Authors · 2025-06-03
>
> Thank you for your review. We address your concerns below.
>
> > The representativeness of the synthetic datasets is limited … conflicts in real applications often arise from nuanced or implicit contradictions rather than explicit textual or visual mismatches.
>
> We agree that synthetic data must be validated against real‑world complexity. In our work, we consider two ways to address cross-modal imbalance. The first is attention manipulation to increase attention on visual inputs. We applied attention manipulation to two multimodal reasoning datasets that demand non-trivial reasoning over the visual domain - popular datasets like MMMU do not show a big gap between text-only and visual models [1].
>
> We see that attention manipulation improves performance by up to 0.8% on HardBLINK, a harder version of the BLINK dataset [1] and 7% on SAT [2]. We provide complete results here: https://docs.google.com/presentation/d/1cW4TGA5kRQjiv7gt8d-Jei3u7EMS01nk24R3kONFpvc/edit?usp=sharing
>
> We also evaluate the effect of finetuning on instance-level mixing datasets. We find that instance-level mixing improves 2.42% and 2.4%. We present complete results in the link above.
>
>
> Overall, we think these experiments show that our proposed methods to improve conflict detection also improve the overall reasoning capabilities of models, especially on datasets that require non-trivial reasoning over each domain. This is really exciting, and we thank the reviewer for the suggestion.
>
> [1] Fu et al. 2024. BLINK: Multimodal Large Language Models Can See but Not Perceive
>
> [2] Ray et al. 2024. SAT: Dynamic Spatial Aptitude Training for Multimodal Language Models
>
> > The proposed instance‑level modality mixing may overfit to the synthetic test format, inflating the apparent gains.
>
> This is a fair concern. We hope our results on attention manipulation and instance‑level modality mixing improving cross-modal reasoning broadly addresses this concern.
>
> Let us know if you have any further questions!

---

> > ### Comment · Reviewer_878T · 2025-06-08
> >
> > Thank you for your detailed response, and it addressed my concerns.

---

### Official Review · Reviewer_93V9 · 2025-05-11

**Rating:** 7
**Confidence:** 5
**Ethics Flag:** 1

**Summary:**

This paper focuses on the cross-modal contextual conflicts in foudation models, including both LLMs and VLMs.
They find that the cross-modal conflict detection performance is much lower than that of the uni-modal one.
They further investigate the causal relationship between the modality-specific attention and the conflict detection performance.
Further, they conclude that the instance-level data mixing significantly improves the ability to detect cross-modal conflicts.

**Questions To Authors:**

1. Can authors explain whether improving cross-modal conflict detection will improve the overall reasoning/understanding capablities of the FMs? For example, you can evaluate the FMs trained on instance-level mixing dataset using MMMU. I think this definitely improves the impact of this work if improving conflict detection relates to improving the overall performance.
2. Since lots of works demonstrate impact of the percentage of the mixing data, I think it is necessary to disentangle this factor from the pipeline. Can authors explain how you consider this factor?
3. There are some missing references on previous knowledge conflict works[1][2][3][4].

[1]  Xie, Jian, et al. "Adaptive chameleon or stubborn sloth: Revealing the behavior of large language models in knowledge conflicts." The Twelfth International Conference on Learning Representations. 2023.

[2] Wang, Yike, et al. "Resolving knowledge conflicts in large language models." arXiv preprint arXiv:2310.00935 (2023).

[3] Su, Zhaochen, et al. "Conflictbank: A benchmark for evaluating the influence of knowledge conflicts in llm." arXiv preprint arXiv:2408.12076 (2024).

[4] Zhu, Tinghui, et al. "Unraveling cross-modality knowledge conflicts in large vision-language models." arXiv preprint arXiv:2410.03659 (2024).

**Reasons To Accept:**

1. This paper discovers a strong causal relationship in the cross-modal/cross-lingual conflict detection, implying that balancing attention bias is the key to conflict detection.
2. The instance-mixing strategy significantly improves the performance of conflict detection, suggesting that the form of multimodal/multilingual instrcution tuning dataset should be improved.

**Reasons To Reject:**

There is only one major weaknesses: the authors are encouraged to increase the credibility of the results.

1. Using LLMs/VLMs to generate textual and visual contents for testing LLMs may create bias in the process. The authors need to conduct sanity check on these generated contents and demonstrate the effectiveness for such a pipeline (i.e. actual conflict rate of the generated contents). Especially, using diffusion models (DALL-E 3) may lead to undesired outputs.
2. Please describe the datasets adopted to fine-tune the FMs: what are they and how they are mixed.

I will raise my score if the authors convince me of the quality of the data.

---

> ### Author Response · Authors · 2025-06-03
>
> Thank you for your detailed review. We address your concerns below.
>
> > Using LLMs/VLMs to generate textual and visual contents for testing LLMs may create bias. The authors need to conduct sanity check on these generated contents and demonstrate the effectiveness.
>
> We agree that a human sanity check is needed, thank you for the suggestion. In each setting, we manually annotated 50 samples and computed the accuracy of our data generation and the LLM-as-a-judge. Here are the results:
> - The cross-lingual QA dataset: 100%
> - The multimodal QA dataset: 96%
> - The monolingual QA dataset: 100%
> - The LM-as-a-judge for LM responses: 96%
>
> We also provide more examples of the generated dataset in https://docs.google.com/presentation/d/1cW4TGA5kRQjiv7gt8d-Jei3u7EMS01nk24R3kONFpvc/edit?usp=sharing
>
> > Please describe the datasets adopted to finetune the FMs: what are they and how they are mixed.
>
> We provided the details in Section 5.1. For multilingual, we used Bactrian-X English and Chinese subsets (Line 266-267); for multimodal, we used the Bactrian-X English subset and visual instruction data from Liu et al. (2024) (Line 270-271). We use pre-existing datasets and just propose a general mixing strategy that can be applied to any dataset. We have added examples of the instance-level mixing dataset in https://docs.google.com/presentation/d/1cW4TGA5kRQjiv7gt8d-Jei3u7EMS01nk24R3kONFpvc/edit?usp=sharing In our current experiments, we used a 1:1 mixing ratio as a proof of concept, without tuning this parameter. We are currently running some ablations with different mixing ratios and hope to report in the next week.
>
> > Can authors explain whether improving cross‑modal conflict detection will improve the overall reasoning/understanding capabilities of the FMs?
>
> This is a great suggestion! In our work, we consider two ways to address cross-modal imbalance. The first is attention manipulation to increase attention on visual inputs. We applied attention manipulation to two multimodal reasoning datasets that demand non-trivial reasoning over the visual domain - popular datasets like MMMU do not show a big gap between text-only and visual models [1].
>
> We see that attention manipulation improves performance by up to 0.8% on HardBLINK, a harder version of the BLINK dataset [1] and 7% on SAT [2]. We provide complete results here: https://docs.google.com/presentation/d/1cW4TGA5kRQjiv7gt8d-Jei3u7EMS01nk24R3kONFpvc/edit?usp=sharing
>
> We also evaluate the effect of finetuning on instance-level mixing datasets. We find that instance-level mixing improves 2.42% and 2.4%. We present complete results in the link above.
>
>
> Overall, we think these experiments show that our proposed methods to improve conflict detection also improve the overall reasoning capabilities of models, especially on datasets that require non-trivial reasoning over each domain. This is really exciting, and we thank the reviewer for the suggestion.
>
> [1] Fu et al. 2024. BLINK: Multimodal Large Language Models Can See but Not Perceive
>
> [2] Ray et al. 2024. SAT: Dynamic Spatial Aptitude Training for Multimodal Language Models
>
> > Impact of the percentage of mixing data; necessary to disentangle this factor
>
> Agreed! We are currently running some ablations with different mixing ratios and hope to report in the next week. We will definitely add it to our final version.
>
> > Missing references
>
> Thank you for the pointers. We will add and discuss them in the final version.
>
> We hope the exact prompt used in dataset constructions, more examples of the dataset and human evaluation have addressed your main concerns about the quality of our dataset. We also hope our results on general reasoning capabilities increase the impact of this work. Please let us know if you have any further questions or suggestions.

---

> > ### Comment · Reviewer_93V9 · 2025-06-08
> >
> > Thanks for the response. I think the authors have addressed the major concerns of the credibility and have made improvements towards greater impacts. I will raise my score.

---

### Official Review · Reviewer_gdZy · 2025-05-12

**Rating:** 6
**Confidence:** 3
**Ethics Flag:** 1

**Summary:**

This paper examines the behavior of language models reasoning over diverse multimodal inputs. In particular, it examines the role of attention across modalities in an attempt to explain and mitigate "attention imbalance" across modalities. They do this in the context of a cross-modal conflict dataset where the different modalities provide conflicting information. They find that in the case of within-modality conflicts LMs do reasonably well at detecting conflicts, while in the case of cross-modality conflicts LMs struggle to detect similar conflicts.

The authors hypothesize that an imbalance in the attention weights across modalities is the problem. To examine this they use a method that manipulates the contributions across modalities by directly changing the relative contributions of modalities. The results appear to confirm the presence of an cross-modal attention imbalance issue.  The paper then presents a modified training regime that mixes cross-modal information at an instance level, demonstrating improved performance on the contradiction detection task.

**Reasons To Accept:**

The overall topic of the reasoning capabilities of cross-modal LLMs is topical and should be of interest to conference participants. Overall the paper is well-organized, motivating the issue, formulating the hypothesis, and presenting experimental results.  The specific experimental evidence for attention imbalance in the context of the data and setup presented in this paper seems compelling. As does the suggested intervention of item-level mixing.  The range of experiments and models deployed is impressive.

**Reasons To Reject:**

* The appendices provide the LM as judge prompt  (" one says something, while the other says something else") but the actual prompt used to generate the contradiction dataset is missing.  In setups like this, a lot depends on the relationship between the prompts used to generate the dataset and the prompt used by the "LLM as a judge".   The specific issue is that the judge prompt is quite vague, and while it arguably includes "conflicting" descriptions, its quite a bit broader than that. In entailment terms, it seems to include contradiction and neutral in the same class. Therefore more detail on the prompt(s) used to generate the datasets is warranted. Is it really the case that what the classifier is doing is actually the task of detecting conflicting data?

* Given the novelty of the task, some level of human quality assessment of the both the generated dataset and the performance of the LLM as judge would strengthen the paper.  In classification tasks like this its entirely possible for there to be artifacts in the data that account for the classifier performance without demonstrating that the classifier is actually doing the task per se.  Similarly, using GPT4o as the data generator and as the judge seems problematic.

* A minor criticism is that its quite a stretch to use the term "multimodal"  to include language/vision and different languages.  I don't think its common practice to consider multilingual language models to be "multimodal" models.

---

> ### Author Response · Authors · 2025-06-03
>
> Thank you for the thoughtful review. Below we respond point‑by‑point.
>
> > The actual prompt used to generate the contradiction dataset is missing.
>
> Here are anonymous links to our prompts used to generate the evaluation datasets (and more examples from each dataset): https://docs.google.com/presentation/d/1cW4TGA5kRQjiv7gt8d-Jei3u7EMS01nk24R3kONFpvc/edit?usp=sharing, and we will add it to the final version. These are the desiderata we went for in our dataset creation, operationalized in the prompts:
> - Diversity of topic coverage
> - Less ambiguity by focusing on facts (e.g., numbers, names, events) instead of opinions
> - No interference with existing knowledge by asking the model to write synthetic news that doesn’t exist
>
>
> > The judge prompt is quite vague, and while it arguably includes "conflicting" descriptions, it’s quite a bit broader than that. In entailment terms, it seems to include contradiction and neutral in the same class.
>
> We would like to point out there are no neutral cases in our dataset by construction. In our generated data, the two pieces of evidence are always contradictory regarding the question (and are consistent in other aspects) and our evaluation measures exactly contradictions. Therefore, a response is considered correct only if it mentions the conflict.
>
> > Given the novelty of the task, some level of human quality assessment of the dataset and of the LLM‑as-a‑judge performance would strengthen the paper. Similarly, using GPT‑4o as both generator and judge seems problematic.
>
> We agree, thanks! In each setting, we manually annotated 50 samples and computed the accuracy of our data and the LLM-as-a-judge. Here are the results:
> - The cross-lingual QA dataset: 100%
> - The multimodal QA dataset: 96%
> - The monolingual QA dataset: 100%
> - The LM-as-a-judge for LM responses: 96%
>
> We also provide more examples of the generated dataset in https://docs.google.com/presentation/d/1cW4TGA5kRQjiv7gt8d-Jei3u7EMS01nk24R3kONFpvc/edit?usp=sharing
>
> We agree that using the same model as both generator and judge creates a potential bias. However, we’d like to highlight that we demonstrate consistent results across multiple open and proprietary models.
>
> > It’s quite a stretch to use “multimodal” for language/vision and different languages.
>
> We will add terms “multilingual” and “cross‑lingual” for the non-visual tasks.
>
> We hope our response provides adequate details about the dataset construction. We will update our manuscript to include prompt descriptions, more details on dataset construction, more examples and  human evaluations. Please let us know if you have any further questions!

---

> > ### Comment · Reviewer_gdZy · 2025-06-08
> >
> > Thank you for the detailed response. The addition of the prompt and the human evaluation of the LLM as a judge accuracy address my most important concerns.

---

### Official Review · Reviewer_BCiN · 2025-06-03

**Rating:** 6
**Confidence:** 3
**Ethics Flag:** 1

**Summary:**

This paper investigates the limitations of foundation models (FMs) in detecting and resolving cross-modal conflicts—situations where different modalities provide conflicting evidence for a given statement.
The authors hypothesize that this ineffectiveness stems from cross-modal attention imbalance, a phenomenon in which FMs assign disproportionately high attention to one modality, which results in neglecting the integration of information across modalities.

To address this issue, the authors propose constructing a new dataset that concatenates instructions from multiple modalities within a single context—a technique referred to as instance-level mixing.
This approach repurposes existing datasets to enhance cross-modal reasoning in FMs.
Training on this newly constructed dataset is shown to improve cross-modal reasoning capabilities and mitigate the problem of attention imbalance across modalities.

**Reasons To Accept:**

- Investigating interactions between modalities in multimodal models remains an active area of research.
- A series of experiments provides empirical and reasonable support for the paper’s claims.
- Although simple, the authors present a practical recipe for mitigating the problem, going beyond merely identifying it.

**Reasons To Reject:**

- There are several minor aspects that could be improved for better presentation. For example, in Figure 3, please specify the label for the y-axis.
- Although the paper presents itself as a multimodal study, it primarily focuses on just two modalities—image and text—making some of the broader claims appear overstated.
- Including multilingual experiments may detract from the central focus of the paper, as the multilingual setting falls under unimodal configurations where only textual input is used.
- Relying mostly on GPT-4o for evaluation carries the risk of weakening the credibility of the findings presented in this work.

---

> ### Author Response · Authors · 2025-06-03
> **Preliminary author response**
>
> Dear reviewer,
>
> Thank you for your thoughtful review. As Phase 1 ends in just a few hours of your review, we are leaving a brief response to keep the discussion open. We will elaborate in a subsequent reply shortly.
>
> 1. We apologize for the confusion in terminology: we believe the same underlying phenomenon explains the results we see in both "multimodal" and "multilingual" settings. We agree that the use of multimodal might be confusing. In our subsequent reply, we will discuss our updated terminology and changes to the paper to clarify this.
>
> 2. We have added new experiments on datasets that require non-trivial reasoning over language and vision such as BLINK [1] and a more recent harder version of BLINK, and SAT [2]. We find that the two methods of addressing cross-modal imbalances (attention manipulation and finetuning with instance-level mixing datasets) provide consistent gains on these real-world difficult benchmarks with gains ranging from 2.5% to 7%. Our complete results on various real-world benchmarks are available here: https://docs.google.com/presentation/d/1cW4TGA5kRQjiv7gt8d-Jei3u7EMS01nk24R3kONFpvc/edit?usp=sharing
>
> [1] Fu et al. 2024. BLINK: Multimodal Large Language Models Can See but Not Perceive
>
> [2] Ray et al. 2024. SAT: Dynamic Spatial Aptitude Training for Multimodal Language Models
>
> 3. We have also added more details about our GPT-4o evaluation dataset construction, including human evaluations and more examples. These are also available at this link: https://docs.google.com/presentation/d/1cW4TGA5kRQjiv7gt8d-Jei3u7EMS01nk24R3kONFpvc/edit?usp=sharing
>
> Thank you once again for your review! We will shortly follow-up with a subsequent response to address the terminology concern in detail with you, along with some more details on the other questions.

---

> > ### Author Response · Authors · 2025-06-08
> >
> > Dear reviewer,
> > To clarify our terminology we will update the terms as follows:
> > -  “Multimodal” for  vision + language inputs
> > - “Multilingual” for the text-only tasks with multiple languages.
> > - “Cross-domain” as an umbrella term we introduce for any setting in which distinct information sources (whether modalities or languages) must be integrated.
> >
> > Although multimodal (vision + language) and multilingual (cross‑lingual, text‑only) inputs differ, our main claim is on the same underlying issue: an attention‑imbalance that harms cross‑domain reasoning whenever the model must reconcile competing information sources, either across languages or modalities. We will highlight this in the introduction and use the clarified terminology consistently throughout the paper.

---

### Decision · Program_Chairs · 2025-07-06

**Decision:**

Accept

**Comment:**

This paper investigates cross-modal contextual conflicts in foundation models (FMs), including both LLMs and VLMs. After the rebuttal, it received scores of 6677. All reviewers responded positively, noting that: (1) the exploration of cross-modal and cross-lingual conflicts is both interesting and insightful, and (2) while the proposed mitigation strategy is simple, it is practical and goes beyond merely identifying the problem. The authors have also committed to including several additional results in the final version of the paper, and it is hoped that these promised additions will be delivered. Overall, the AC recommends acceptance.